# Effect of the Supplementation Using an Herbal Mixture as a Choline Source during Early Gestation in Rambouillet Ewes

**DOI:** 10.3390/ani13040645

**Published:** 2023-02-13

**Authors:** Nydia Emilce Suárez-Suárez, Héctor Aaron Lee-Rangel, Augusto César Lizarazo-Chaparro, German David Mendoza-Martínez, Guillermo Espinosa-Reyes, Pedro Abel Hernández-García, Juan Carlos García-López, José Antonio Martínez-García, Gregorio Álvarez-Fuentes, José Alejandro Roque-Jiménez

**Affiliations:** 1Facultad de Agronomía y Veterinaria, Universidad Autónoma de San Luis Potosí, San Luis Potosí 78321, Mexico; 2Centro de Enseñanza Práctica e Investigación en Producción y Salud Animal, Universidad Nacional Autónoma de México, Mexico City 14500, Mexico; 3Departamento de Producción Agrícola y Animal, Universidad Autónoma Metropolitana-Xochimilco, Ciudad de México 04960, Mexico; 4Facultad de Medicina-CIACYT, Universidad Autónoma de San Luis Potosí, San Luis Potosí 78220, Mexico; 5Centro Universitario UAEM Amecameca, Universidad Autónoma del Estado de México, Amecameca 56900, Mexico; 6Instituto de Investigación de Zonas Desérticas, Universidad Autónoma de San Luis Potosí, San Luis Potosí 78377, Mexico

**Keywords:** offspring, fetal programming, milk, colostrum, ruminant, choline

## Abstract

**Simple Summary:**

Early gestation is a critical period where the establishment of correct placentation is one of the key steps to ensuring a successful pregnancy outcome. Choline is an essential nutrient for cell membranes’ structural integrity, signaling functions, and central fetus nervous system development. Different studies have reported that choline plays a critical role during late gestation by increasing colostrum and milk quality and improving newborn mammal development. However, there is a lack of studies that describe the role of choline when the placenta requires to potentialize its nutrients’ supply to the embryo and to potentialize the fetal development that increases the ability of growth during early life. The current article aims to describe the positive effects of offspring development from ewes supplemented during early gestation using an herbal mixture as a choline source.

**Abstract:**

Previous research indicates that adequate choline nutrition during late gestation improves fetal development. However, there is a lack of studies describing choline’s role during early gestation. Thus, the current study hypothesizes that an herbal mixture as a source of choline (Biocholine) positively affects offspring development from ewes supplemented during early gestation. Therefore, the objectives were to evaluate the impact of biocholine on the programming of the offspring early in life through the evaluation of dams and newborn performance. Twenty-eight four-year-old Rambouillet ewes were assigned randomly to two treatments: non-supplementation and 4 gd^−1^ of biocholine during the early gestation. Compared with the dams without supplementation, the ewes supplemented using biocholine showed no increase in parameters such as birth and weaning weight (*p* > 0.05). Additionally, the milk yield and quality of colostrum and milk did not present statistical differences (*p* > 0.05). However, the placental membrane development was reduced in the ewes that received supplementation with biocholine; interestingly, those dams increased the weight of the newborns during the lambing period (*p* < 0.05). Finally, the current study proposes necessary elucidation of how placental size is programmed and if less placental development has potential benefits in the fetus’s development.

## 1. Introduction

In recent years, one of the significant problems in livestock system production is the unfavorable geological locations that result in forage deficiency during gestational stages [1]. This challenge involved poor maternal nutrition changes in the physiological and metabolic phenotypes of the offspring [2], consequently adapting the fetus during its developmental [3]. The early gestation period is critical when nutrition could start programming the offspring [4]. The programming begins with conception, zygote, morula, blastocyst, embryo, and fetus development [5], and with the formation of the first organs, such as the central fetal nervous system, liver, heart, and digestive system [6]. However, Larqué et al. [5] and Roque-Jimenez et al. [6] reported that maternal nutrition results during gestation begin with the offspring’s growth and might be confounded or related to postnatal development during the lactation period. One reason could be that mammary gland function and colostrum yield could also be impacted by nutritional management prepartum, where proteins, amino acids, fatty acids, or vitamins can modify different metabolic pathways [7].

In recent years, researchers have established that vitamins are essential for mammals when a sufficient supply of amino acids is unavailable in the diet [8]. For this reason, new reports described that folate, vitamin B, or a precursor of these vitamins, such as choline, could be involved in the nutrient supply to the fetus. According to Pinotti et al. [8], mammals’ choline has been classified as one of the B-complex vitamins, but it does not satisfy the standard definition of a vitamin. In ruminants, choline is described as a methyl donor and a lipotropic agent [9]. Moreover, in dairy cows, Swartz et al. [10] described that choline might make the remethylation of homocysteine via its metabolite betaine and be the spare crucial metabolite in mammals. Accordingly, choline plays a vital role in ruminants’ energy and protein metabolism [8]. The use of rumen-protected choline has demonstrated that choline is a limiting nutrient for milk production in dairy cattle [11]; nevertheless, evaluations of the herbal mix as a source of choline in ruminants have indicated that some of these products have rumen-protected nutrients that are absorbed as choline promotors recycling amino acids and vitamins used by the animal system when choline is limited in the diet [12]. Such blocking or potentializing of the metabolic ways is related to the bioactive compounds that contain herbal mixes that could include fatty acids, thymol, polyphenols, isothiocyanates, saponins, and terpenoids [12,13], which benefit the recycling of amino acids, vitamins, or fatty acids [14].

Herbal mixes have been tested in different animal models as sources of vitamins and amino acids. Crosby et al. [15] reported that the ewes supplemented with 4 gd^−1^ of an herbal mix as a source of choline from 30 days before to 30 days after birth increased lamb birth weight, milk yield, and the oleic acid profile in the milk. Additionally, supplementing 4 gd^−1^ of the herbal mix as a source of choline to lambs during the final period benefits the mobilizing of non-esterified fatty acids, stimulating glucose and cholesterol synthesis [16]. Nevertheless, despite all recommendations on choline intake in mammals, there needs to be more research on the use of sources of choline in early gestation in domestic animals. 

Early gestation is a critical period where the establishment of correct placentation is one of the key steps to ensuring a successful pregnancy outcome [17]. As the primary factor of the fetal period, the placenta is the first organ that forms during embryonic development [5]. It modulates the fetus-dam interface, whose primary function is to regulate the exchange of respiratory gases, nutrients, and waste products [17,18]. Most studies that describe the effect of embryo development with choline sources of supplementation during early gestation are focused on humans [5]. However, the research is focused on explaining the availability of choline and acetyl-CoA influence on choline acetyltransferase activity in the fetal central nervous system; but it is less concerned with placental characterization or the effect in the storage of choline metabolites that may transfer to offspring during the lactation period [19]. Thus, and based on the previous background, we hypothesized that herbal mix as a source of choline positively affects the developmental parameters of offspring from ewes supplemented during early gestation. Accordingly, the objectives were to evaluate and describe the effect of herbal mix supplementation on the programming of the offspring early in life through the evaluation of the colostrum and milk quality.

## 2. Materials and Methods

### 2.1. Ethics

The Committee of Animal Care reviewed the procedures on ewes and offspring based on the Ethical for Animal Procedures (Approval Code: DCA-1302/2018). The Universidad Autónoma de San Luis Potosí approves this Committee, and its decisions are established according to requirements by the Mexican government and current Federal Laws on technical specifications for the care and use of laboratory animals and for livestock farms (NOM-062-ZOO-1995) [20]. Additionally, the ewes and lambs were cared for following the procedures proposed by Menzies et al. [21] for Lambing Management and Neonatal Care.

### 2.2. Location

The current research was conducted at the Sheep Center of the Facultad de Agronomía y Veterinaria, established in Soledad de Graciano Sánchez, San Luis Potosí, México (Latitude 22°14′0.58″; Longitude 100°50′48.5″), from October 2021 to April 2022. The breed of the flock at the sheep center is Rambouillet.

### 2.3. Animals, Treatments, and Sampling

#### 2.3.1. Selection of Ewes

Twenty-eight four-year-old Rambouillet ewes with previously two lactations (53 Kg ± 0.5 initial body weight [IBW]) were randomly selected from a group of fifty ewes one month before mating. During this month, the chosen ewes were revised using a clinical examination that involved the interpretation of their clinical historical information in the flock within the farm environment.

#### 2.3.2. Mating

For mating, the ewes were divided into two groups (14 ewes) and housed in two pens (10 m in length, 10 m in width, and 2.50 m in height with their roof sheet). Later, in each pen, a ram was introduced (IBW 80 Kg ± 0.5) and equipped with a marking harness with different paint colors to mark the wool of the ewes during the mating. The total mating time was four consecutive days. When the ram marked each ewe after mating, the ewe was considered mated. Later, the ewes with color marks were housed in an individual pen and randomly assigned to a treatment.

Thirty days after the mating, ewes were tested by a noninvasive imaging test to confirm the pregnancy stage (Ultrasound V1; Xuzhou Kaixin Electronic, Xuzhou, China). The ultrasonography confirmed a 100% pregnancy rate. Thus, all ewes received the supplementation during the first third of the gestation period.

#### 2.3.3. Supplementation and Gestation

The day of the paint mark was regarded as the day of mating/conception (day 1). The early gestation supplementation for the current study was considered from day 1 (mating/conception) to day 51 (the last day of the first third of gestation). After mating, the twenty-eight pregnant ewes (fourteen ewes for treatment) were randomly assigned to the two treatments: (control) not-supplementation of choline from an herbal mix source; and supplementation with 4 gd^−1^ of the herbal mix (*Achyranthes aspera*, *Trachyspermum ammi*, *Azadirachta indica*, *Citrullus colocynthis*, *and Andrographis paniculate*) as a source of choline (Biocholine; Nuproxa^®^, Etoy, Switzerland). 

During the entire gestation period, the ewes were housed in individual pens. The sizes and equipment of the individual pens were designed according to ewes and nursing lambs [21]. The individual pen dimensions were 2.5 m in length, 2.5 m in width, and 1.50 m in height with their roof sheet. The walls for the pens were welded mesh equipped with a canvas cover to reduce the impact of the dominant air. The floor was provided with dry straws, shavings, and sawdust. Each pen was equipped with a feeder, drinker for the dam, and drinker for the lamb. 

The herbal mix supplementation was top-dressed on the feed using one gd^−1^ of sugar molasses as a carrier to asseverate de entire intake of the herbal mix dust. During all experiment periods, the ewes were fed a mixture of forages chopped of 50% alfalfa and 50% oat ad libitum. The dose was designed based on previous results. Roque-Jimenez et al. [12] reported that this dose contains 0.0725 gd^−1^ choline conjugates, an amount recommended in mammals during the gestation period [19]; also, other studies reported that this dosage increases epigenetic marks, lamb birth weight, milk yield, and decreases disease incidence in ruminants [12,15,22].

#### 2.3.4. Births, Breeding, and Sampling

All the ewes had single births. Additionally, the ewes were given birth under similar weather (temperature averages 28 °C maximum and 20 °C minimum). All deliveries were normal lambing. The ewes were watched during the 24 h by the staff of the sheep center ten days before the date of the establishment for the lambing period of the group of ewes. During births, the expulsion time was recorded from the first waterbag into the cervix ruptured to release the amniotic fluid until the entire placenta tissue expulsion. After delivery, each ewe took care of her newborn lambs while the ewes expulsed their placenta in its totality naturally. Immediately after the placenta was expelled from the ewe, the staff of the sheep center removed it from the pen using a hook so as not to interfere with the ewe’s claim of her lamb and allow her to nurse. Finally, the entire placenta tissue was weighed using a digital weight scale for the weight recording (Rhino^®^, Ciudad López Mateos, Mexico).

The developmental test for the offspring followed the procedure by Roque-Jimenez et al. [12], where the offspring were weighed using a steel sheep weighing scale (Begame-800, Rhino^®^, México) on day 1 (lambing) and 5 weeks later (35 days old). Daily weight gain (DWG) was calculated for days 1 to 35. The milk yield was measured each week following the methodology by Reynolds et al. [23] until day 35 (5 weeks of sampling). Briefly: ewes need to be separated from their offspring and immediately milked by hand; the keepers offer this milk to the newborn lambs. After three hours, an injection of oxytocin (20 IU) into the jugular vein was applied, waiting five minutes to begin hand milking; the yield (mL) was measured in a graduated cylinder and recorded by the milk sample collected. These colostrum and milk samples were frozen at −20 °C until further analysis of colostrum and milk quality.

Each colostrum sample was put in a water bath for 1 min (35 °C), homogenized by a vortex (Vortex-Genie, Ocala, FL, USA) until the colostrum temperature reached 29 °C, and analyzed according to the manufacturer’s protocol for colostrum using the Lactoscan Ultrasonic Milk Analyzer (Milkotronic, Nova Zagora, Bulgaria). For the milk analysis, the procedure was identical to the colostrum analysis. Each milk sample was previously mixed and homogenized in a water bath for 1 min (35 °C) until the milk temperature reached 29 °C; after this, the samples were analyzed according to the manufacturer’s protocol for milk in the same equipment. 

It is essential to highlight that none of the samples were diluted for the colostrum and milk analysis. Additionally, it is remarkable that all the lambs survived the entire experimental period.

### 2.4. Statistical Analysis

The experimental design was a completely randomized design. The data were analyzed with the MIXED procedure of SAS (9.4 SAS Inst. Inc., Cary, NC, USA), where the ewes were a random component, and the treatments were the fixed components in the model. The PDIFF option of SAS was used for mean separation. A probability of ≤0.05 was considered statistically significant.

## 3. Results

### 3.1. Productive Performance

There were no statistically significant differences in the productive performance of the Rambouillet ewes (*p* > 0.05). Furthermore, there were no statistically significant differences (*p* > 0.05) among the treatments for milk yield (Table 1).

### 3.2. Colostrum and Milk Quality

There was no difference between the treatments for the quality of colostrum and milk (*p* > 0.05) (Table 2).

### 3.3. Placental Weight and Time of Membrane Expulsion

There was a significant effect (*p* < 0.05) of biocholine supplementation on the weight of the placental membrane at the moment of lambing (Table 3). 

### 3.4. Offspring Development

The lambs from ewes supplemented with biocholine during early gestation showed greater weight than those born from dams without supplementation (*p* = 0.02) (Table 4).

## 4. Discussion

These results are similar to the data presented by Roque-Jimenez et al. [12], where no differences were observed in the birth weight, initial weight, final weight, and live weight changes in the ewes supplemented with 4 g^−1^ of biocholine during the entire gestation. However, extensive reports describe those herbal mixes as choline sources, or that rumen-protected choline has positive effects on fetal growth and milk production, but only during supplementations in the last third of gestation in ewes and dairy cows [10,15,24]. The differences in milk yield could be explained by the moment of the supplementation in the different thirds of gestation, according to which of three-thirds of the pregnancy receives supplementation with nutrients used in critical moments by the placenta, fetus, or mammary gland [25]. According to Du et al. [1], the intake of nutrients by the dam during the first third of gestation may be used for the placenta and fetus development with epigenetic biomarkers until the fetus grows; this is because myogenesis, adipogenesis, and fibrogenesis in the fetus require a greater amount of metabolites that can contribute to more desirable phenotypes and establish phenotype inheritance across generations [1]. Interestingly, reviews and research postulated the description of supplementation with amino acids or vitamins using genomic analysis in the last two years. The authors concluded that the nutrients with a potential role in methyl donation are used immediately by the mother for diverse biological functions such as fetal development, placenta growth, immune modulation, DNA methylation, vascularization, and organogenesis [12,26], and not precisely modifying the growth or body condition of the dams as in the current report [27]. However, these theories are not included in our study due to the lack of epigenetic biomarkers tests.

In ruminants, colostrum and milk have been described as a secretion synthesized by the mammary gland during the last third of the gestational period [12]. Moreover, colostrum and milk are accumulated during the lambing period, and their quality depends on the dam’s diet [28]. Nevertheless, the percentages of milk and colostrum quality values of our study are similar to other studies where the offspring showed positive performance and good daily weight gain [12,29]. The placental weight was lightest in ewes supplemented with biocholine. This result contrasts with most reports that describe the placenta’s development as requiring the most considerable weight to produce an effect on the nutrient supply to the fetus [30]. Interestingly, Magolski et al. [31] reported that the breed specialists of ewes used in wool production stated that the ewes required nutritional supplements either during conception or during the critical period of fetal development, where the placental development growth increase accelerated, and the fetus increased the nutrient demand. Moreover, some studies on ewes have shown that nutrient restriction affects placental and fetal growth [32,33]; however, our study has demonstrated that ewes with the lightest placental weight breed the lambs with the most significant weight. 

We cannot explain the lightest weight in placental membranes from ewes supplemented with biocholine during early gestation due to the lack of reports with similar results. In ruminants, including sheep, placenta size has been previously used as an indirect measurement of nutrient delivery to the fetus [30,34]. Nevertheless, King et al. [35] and Kwan et al. [36] used a different mammalian model (mice), and they concluded that choline supplementation modulated the different ways that the placental nutrients were transported to the fetus. In beef cattle, Camacho et al. [37] demonstrated that the umbilical cord increases the umbilical blood flow affecting the fetal growth trajectory and reducing intrauterine growth restriction, miscarriage, and preeclampsia; however, they concluded that independent of the size of the placenta, the offspring modulate its growth by the nutrients and not for the placental membrane development. Thus, the conclusions by Camacho et al. [36] are comparable to our results, where the weight of the lamb’s newborn is greater than the control offspring, but they are from the lightest placenta [37,38]. Additionally, other investigations have suggested that choline may increase the number of caruncles and cotyledons in the placental tissue [30,39]. We theorized that this increase in the number of sides in the placenta circulation might stimulate the blood flow in the uterine artery but not precisely increase the weight of the total placental membrane. However, further research is necessary to elucidate how placental size is programmed when the requirements of the nutrients are not limited or when the offspring’s weight during birth is not considered low.

What needs to be discovered from the current experiment is whether the changes in the placenta weight observed in lambs born from dams supplemented with biocholine during the first third of gestation positively affected the fetal size and growth. As described previously, the herbal source used in the current experiment has been described using extraction of a bioactive procedure coupled to gas chromatography with mass spectrophotometry by Roque-Jimenez et al. [12]. Interestingly, most of the biocompounds found were methyl groups. According to Estrada-Cortés et al. [39], providing the methyl-donor choline to the preimplantation embryo could positively alter its developmental program to increase birth weight and postnatal changes in DNA methylation patterns in muscle, especially for genes related to anabolic processes and cellular growth. 

Thus, although developed based on a small sample set of ewes and their offspring, the association we found in this study suggests that choline supplementation during the first third of gestation requires further investigation with an affinity to epigenetic modification as gene expression or DNA methylation. 

## 5. Conclusions

The current study allows us to detect the effects of an herbal mix as a choline source included during the first third of gestation on purebred Rambouillet ewes. Significant changes in the lightest placental membranes from ewes supplemented using the herbal source of choline could be used to describe the bioactive and phytogenic compounds, with a possible role in adaptation to pregnancy and fetal development. The supplementation with this herbal mixture as a choline source impacted the birth weight in offspring. Still, interestingly it did not increase the milk yield and some quality parameters of colostrum and milk. Future research should focus on elucidating the relationship among the varying intake patterns of herbal formulas during pregnancy, the epigenetic biomarkers effects, and the subsequent health outcomes in adult life to increase the information on the role of bioactive and phytogenic compounds during placental membrane growth and fetus development.

## Figures and Tables

**Table 1 animals-13-00645-t001:** Productive performance of Rambouillet ewes supplemented with an herbal mix as a source of choline during early gestation.

Item	Treatments		SEM ^1^	*p*-Values
	Control	Biocholine		
Item				
Lactating Ewe weight (Kg)				
At parturition	56.55	62.35	5.29	0.63
At weaning (35-d)	54.78	54.05	5.44	0.49
Total weight change (Kg)	4.43	6.3	0.01	0.47
Daily weight changes (Kg)	0.07	0.105	0.01	0.47
Average daily milk production (mL)	681	690.8	31.98	0.65

^1^ SEM: standard error of the mean.

**Table 2 animals-13-00645-t002:** Effect on the quality of colostrum and milk from ewes supplemented with an herbal mix as a source of choline during early gestation.

Item	Treatments		SEM ^1^	*p*-Values
	Control	Biocholine		
Colostrum quality, %				
Fat	9.54	11.67	1.59	0.36
Protein	6.84	5.54	0.51	0.11
Solids	18.66	13.82	1.68	0.08
Lactose	10.25	8.21	0.92	0.14
Total solids	26.64	25.44	2.01	0.67
Density (Kg/m^3^)	1062.24	1044.01	6.37	0.07
Milk quality, %				
Fat	6.77	6.45	0.50	0.66
Protein	3.44	3.30	0.09	0.30
Solids	9.37	9.07	0.24	0.41
Lactose	5.15	4.98	0.13	0.74
Total solids	15.37	14.75	0.49	0.39
Density (Kg/m^3^)	1029.65	1028.51	1.48	0.49

^1^ SEM: standard error of the mean.

**Table 3 animals-13-00645-t003:** Placental weight and time of placental membrane expulsion during lambing from ewes supplemented using an herbal mix as a source of choline during early gestation.

Item	Treatments		SEM ^1^	*p*-Values
	Control	Biocholine		
Retention time (min)	169.60	194.50	7.12	0.23
Placental weight (g)	451.21	377.10	32.9	0.05

^1^ SEM: standard error of the mean.

**Table 4 animals-13-00645-t004:** Productive performance of lambs born from Rambouillet ewes supplemented with an herbal mix as a source of choline during early gestation.

Item	Treatments		SEM ^1^	*p*-Values
	Control	Biocholine		
Birth weight (Kg)	5.25	6.16	0.67	0.02
Weaning weight (Kg)	21.73	20.38	2.87	0.47

^1^ SEM: standard error of the mean.

## Data Availability

The data presented in this study are available from the corresponding author upon request. The data are not publicly available due to institutional instructions.

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
