# Peer review of "Effect of the Supplementation Using an Herbal Mixture as a Choline Source during Early Gestation in Rambouillet Ewes"

_animals, 2023, doi:10.3390/ani13040645_

Round 1

Reviewer 1 Report

All opinion was emitted with all respect to the efforts of the authors for the preparation of the experiment and its report

The purpose of this study was to evaluate the effects of the supplementation using an herbal mixture as a choline source during early gestation in Rambouillet ewes (positive effects of supplementation on late period of gestation have been previously observed) on weight changes of ewes and milk yield and quality, as well as the weight change of lambs and placental mass weight. To accomplish the above, researchers applied two treatments (0 supplemental, and 4 g supplemental herbal/d) to 28 ewes in the first 51 days of gestation. I think that inclusion of 4g supplementation treatment during the final phase of the gestation period would have been interesting to consider (to directly contrast the supplementation method). And given the results obtained with the placental mass, a histological study could have provided additional information. Even so, this manuscript reports a topic pertinent to contemporary readers. The manuscript is well written and organized. The data presented are good and sufficient to provide new information to contrast the hypothesis raised. Justification of the experiment and discussion is well supported however, there are few little flaws mainly in the Mat & Met section (listed below) which in my opinion should be rectified before the paper is accepted to be published in Animals journal.

 L124: Please, describe characteristics of pens.

L125: noninvasive imaging test (Indicate here the model of ultrasound equipment used, i.e. Mindray Vetus EQ7 Ultrasound Machine,Duluth, Georgia, USA ) to confirm..

L126: Rewrite as: The ultrasonography confirmed a 100% pregnancy rate

L129: Please indicate the source of herbal mix sources used, its presentation (powder, liquid) and its choline conjugated concentration expected per g of herbal mix.

L131: How did you top-dress the 4 g of herbal mix on the diet? Used any carrier? Herbal mix was sprayed in the diet? Please, specify

L132: Please indicate the proportion in mixture of each forage offered (i.e., in a 60:40 ratio)

L139-150: Describe here how do you handle placentas for weighing

L147: Type of scale used for weighing ewes and lambs and how did you measure the milk yield?

L150: Following the methodology by Reynolds et al [22]. Briefly: ewes need to be

Table 1. Needs to be more properly described. For example

Item

Weight (kg)

    At parturition

    At weaning (35-d)

 Total weight change (kg)

 Daily weight change (%)

 Average daily milk production (mL)

Table 2

I have a doubt about colostrum density values expression. Please check (to my knowledge, the normal density ranges are around of 1.030 to 1.040 mg/mL)

Author Response

Reviewer #1

All opinion was emitted with all respect to the efforts of the authors for the preparation of the experiment and its report

AR: We appreciate your comments and recommendations for improving the manuscript's quality. The changes were highlighted using the tracking changes to mark the editions on the manuscript.

The purpose of this study was to evaluate the effects of the supplementation using an herbal mixture as a choline source during early gestation in Rambouillet ewes (positive effects of supplementation on late period of gestation have been previously observed) on weight changes of ewes and milk yield and quality, as well as the weight change of lambs and placental mass weight. To accomplish the above, researchers applied two treatments (0 supplemental, and 4 g supplemental herbal/d) to 28 ewes in the first 51 days of gestation. I think that inclusion of 4g supplementation treatment during the final phase of the gestation period would have been interesting to consider (to directly contrast the supplementation method). And given the results obtained with the placental mass, a histological study could have provided additional information. Even so, this manuscript reports a topic pertinent to contemporary readers. The manuscript is well written and organized. The data presented are good and sufficient to provide new information to contrast the hypothesis raised. Justification of the experiment and discussion is well supported however, there are few little flaws mainly in the Mat & Met section (listed below) which in my opinion should be rectified before the paper is accepted to be published in Animals journal.

 L124: Please, describe characteristics of pens.

AR: The concern was addressed and changed. We added information about pen sizes and equipment. Line 125 to 142.

L125: noninvasive imaging test (Indicate here the model of ultrasound equipment used, i.e. Mindray Vetus EQ7 Ultrasound Machine,Duluth, Georgia, USA ) to confirm..

AR: The concern was addressed and changed. We added information about de model and type of ultrasound. Line 137.

L126: Rewrite as: The ultrasonography confirmed a 100% pregnancy rate

AR: The concern was addressed and changed.

L129: Please indicate the source of herbal mix sources used, its presentation (powder, liquid) and its choline conjugated concentration expected per g of herbal mix.

AR: The concern was addressed and changed. Line 143 to 144.

L131: How did you top-dress the 4 g of herbal mix on the diet? Used any carrier? Herbal mix was sprayed in the diet? Please, specify

AR: We appreciate the reviewer’s comment. We added information in lines 143 and 144. Also, we edited lines 140 to 142 to increase the description of the vegetal species used in the herbal mix.

L132: Please indicate the proportion in mixture of each forage offered (i.e., in a 60:40 ratio)

AR: We appreciate the reviewer’s comment. We added information in lines 145 and 146.

L139-150: Describe here how do you handle placentas for weighing

AR: We appreciate the reviewer’s comment. We added information in lines 153 to 161.

L147: Type of scale used for weighing ewes and lambs and how did you measure the milk yield?

AR: We appreciate the reviewer’s comment. We added information in lines 168 to 172.

L150: Following the methodology by Reynolds et al [22]. Briefly: ewes need to be

AR: We appreciate the reviewer’s comment. We edited the description and increased the details to better comprehension.

Table 1. Needs to be more properly described. For example

Item

Weight (kg)

    At parturition

    At weaning (35-d)

 Total weight change (kg)

 Daily weight change (%)

 Average daily milk production (mL)

AR: We appreciate the comment. The concern was addressed and changed.

Table 2

I have a doubt about colostrum density values expression. Please check (to my knowledge, the normal density ranges are around of 1.030 to 1.040 mg/mL)

AR: The description by the reviewer is correct. We interpretedwrong the density data observed in the lactoscan milkotronic analyzer. The manual of the manufacturer describes:

** Density data are shown in an abbreviated form. For example, 27.3 have to be understood as 1027.3 kg/m3. To determine the milk density, write down the result from the display and add 1000.  Example: result 21,20; density = 1000 + 21,20 = 1021,2 kg/m3.

Thus, we edited the data and the table.

We appreciate this observation.

Reviewer 2 Report

This is an interesting paper which provides added knowledge regarding Choline's impact on placental development, colostrum & milk production, as well as lamb growth. I think this is an interesting area of research and seems to be an area which could utilize more investigation.

There seems to be more materials and methods which are not fully explained in the paper, that I believe need to be more clearly stated. Specifically:

-Is the herbal mix commercially available or what exactly is the source / product used?

- When were treatments assigned? This is confusing in the paper (lines 120-136)

- How were the ewes housed between d 51 of gestation to lambing?

- Pen sizes during the project?

- How were plancentas handled?

- How many lambs did the ewes have?

- What were milk & colostrum samples analyzed for?

I noticed in Table 4 there were numerical weight differences at 35 days of age, where the lambs heavier at birth (biocholine treatment) were lighter, do you have any thoughts on this observation?

On Line 86, I believe that should read "non-esterified"

Author Response

Reviewer #2

This is an interesting paper which provides added knowledge regarding Choline's impact on placental development, colostrum & milk production, as well as lamb growth. I think this is an interesting area of research and seems to be an area which could utilize more investigation.

There seems to be more materials and methods which are not fully explained in the paper, that I believe need to be more clearly stated. Specifically:

-Is the herbal mix commercially available or what exactly is the source / product used?

AR: We appreciate the reviewer’s comment. We added information in lines 140 and 141.

- When were treatments assigned? This is confusing in the paper (lines 120-136)

AR: We appreciate the reviewer’s comment. We edited the entire paragraph to increase the comprehension of the procedure.

- How were the ewes housed between d 51 of gestation to lambing?

AR: We appreciate the reviewer’s comment. We added information in lines 129 to 142.

- Pen sizes during the project?

AR: We appreciate the reviewer’s comment. We added information on pen size in lines 125 to 142.

- How were plancentas handled?

AR: We appreciate the reviewer’s comment. We added information in lines 153 to 164.

- How many lambs did the ewes have?

AR: We appreciate the reviewer’s comment, line 154 was edited with this information.

- What were milk & colostrum samples analyzed for?

AR: We appreciate the reviewer’s comment. We added a new paragraph to the better description for future analysis after preserving colostrum and milk. Line 173 to 180.

I noticed in Table 4 there were numerical weight differences at 35 days of age, where the lambs heavier at birth (biocholine treatment) were lighter, do you have any thoughts on this observation?

AR: We appreciate the reviewer’s comment. Our group has published different studies using herbal formulas. We theorized that this numerical difference is related to the compounds as metabolites or bio-compounds that increase the effort of the offspring to grow in epigenetic ways, increasing metabolic pathways for adult life. These modifications may not be related to growth performance in the offspring during weaning.

Few articles are established that correlated the consumption of bio-compounds from plant extracts such as oils and herbal formulas in pregnancy with offspring growth.

O’Neill et al. indicated that supplementation during pregnancy with herbal formulas or plant extracts might lead to epigenetic effects across target organs, tissues, whole blood, and or cell types in the offspring; however, O’Neill et al. and McGee et al. suggested that although the herbal mix supplementation may not impact offspring birth weight, it could alter gene regulation and decrease the risk of metabolic diseases during early life and adulthood.

On Line 86, I believe that should read "non-esterified"

AR: We appreciate the comment. The concern was addressed and changed.

Round 2

Reviewer 1 Report

I have read the revised manuscript and appreciate the authors' consideration of my previous suggestions. I have no further review comments. Respectfully, I consider that paper now meets the quality requirements to be published

Author Response

Reviewer #1

I have read the revised manuscript and appreciate the authors' consideration of my previous suggestions. I have no further review comments. Respectfully, I consider that paper now meets the quality requirements to be published.

AR: We appreciate your comments and recommendations for improving the manuscript's quality.

Reviewer 2 Report

Some of the materials and methods are still confusing, especially the Animals & Treatments. Specifically, Lines 120 - 140, where it still seems like treatments are assigned prior to mating or maybe after pregnancy confirmation??

Did all ewes have single lambs? This is not clear as current written currently. Line 154, "All the ewes had one rearing during the births", this sentence does not make sense to me, can you please explain what is meant by it?

How long after delivery was the placenta weighed?

Was the milk production measured weekly collected via a different procedure than the milk yield on d 35? Were these weekly samples used to determine daily milk production represented in Table 1? If so, this is needs to be stated in the Materials & Methods (maybe around Line 165 or so).

Line 263, the Camacho study utilized Beef cattle, not Dairy cattle as stated.

Double check the citations, as on line 283 you list a reference # 40, whereas the references end at # 39.

Author Response

Reviewer #2

Some of the materials and methods are still confusing, especially the Animals & Treatments. Specifically, Lines 120 - 140, where it still seems like treatments are assigned prior to mating or maybe after pregnancy confirmation??

AR: We appreciate your comments and recommendations for improving the manuscript's quality. We added subscripts to increase the description of the animals, treatments, and sampling sections. The report was divided into information about how the ewes were divided, mating, and later kept in individuals' pens. Also, the reviewer's concern was addressed about how the ewes started their supplementation after mating. Lines 120 to 197.

Did all ewes have single lambs? This is not clear as current written currently. Line 154, "All the ewes had one rearing during the births", this sentence does not make sense to me, can you please explain what is meant by it?

AR: We appreciate your comments and recommendations for improving the manuscript's quality. We added subscripts to increase the description of the number of births per ewe. Line 168. We appreciate this comment and edited this paragraph for a better description.

How long after delivery was the placenta weighed?

AR: We appreciate your comments and recommendations for improving the manuscript's quality. We added subscripts (2.3.4 Births, breeding, and sampling) to increase the description of how the placental tissue weight has been recorded. Additionally, we edited and advanced the explanation of how the staff of the sheep center managed the births, with remarkably that all placental expulsion was natural.

Was the milk production measured weekly collected via a different procedure than the milk yield on d 35? Were these weekly samples used to determine daily milk production represented in Table 1? If so, this is needs to be stated in the Materials & Methods (maybe around Line 165 or so).

AR: We appreciate your comments and recommendations for improving the manuscript's quality. We added subscripts (2.3.4 Births, breeding, and sampling) to a better description of the milk yield that was measured.

Line 263, the Camacho study utilized Beef cattle, not Dairy cattle as stated.

AR: The concern was addressed and changed.

Double check the citations, as on line 283 you list a reference # 40, whereas the references end at # 39.

AR: We appreciate your comments and recommendations for improving the manuscript's quality. The concern was addressed and changed.
